# Is Foundational Movement Skill Competency Important for Keeping Children Physically Active and at a Healthy Weight?

**DOI:** 10.3390/ijerph19010105

**Published:** 2021-12-23

**Authors:** Jonathan D. Foulkes, Zoe Knowles, Stuart J. Fairclough, Gareth Stratton, Mareesa V. O’Dwyer, Lawrence Foweather

**Affiliations:** 1School of Sport & Exercise Sciences, Liverpool John Moores University, Student Life Building, Copperas Hill, Liverpool L3 5AJ, UK; 2Physical Activity Exchange, Research Institute for Sport & Exercise Sciences, Liverpool John Moores University, 5 Primrose Hill, Liverpool L3 2EX, UK; Z.R.Knowles@ljmu.ac.uk (Z.K.); L.Foweather@ljmu.ac.uk (L.F.); 3Movement Behaviours, Health and Wellbeing Research Group, Department of Sport & Physical Activity, Edge Hill University, St Helens Road, Ormskirk L39 4QP, UK; Stuart.Fairclough@edgehill.ac.uk; 4College of Engineering, Swansea University, Swansea SA2 8PP, UK; g.stratton@swansea.ac.uk; 5Research Institute for Sport & Exercise Sciences, Liverpool John Moores University, Liverpool L3 3AT, UK; modwyer@betterstart.ie

**Keywords:** foundational movement skill, fundamental movement skills, physical activity, preschool, primary, weight status, longitudinal

## Abstract

This longitudinal study examines the associations between foundational movement skills (FMS) competency, moderate-to-vigorous physical activity (MVPA) and weight status among children (*n* = 75) attending preschools in deprived areas from early to late childhood. Twelve FMS were assessed using the Children’s Activity and Movement in Preschool Motor Skills Protocol and video analysis. Physical activity was measured via hip-mounted accelerometry. Data was collected over a five-year period, with Baseline Follow Up data collected between 2010 and 2015. There was an overall pattern of increase for total, object-control and locomotor scores between Baseline and Follow-Up. Conversely, there was an overall pattern of decline for MVPA among participants. There was a positive significant (*p* < 0.05) association between total and locomotor scores and MVPA at Baseline. However, these associations weakened over time and no significant associations were found at Follow-Up. Baseline competency failed to predict Follow-Up MVPA or weight status. Likewise, Baseline MVPA was not found to be a predictor of Follow-Up FMS competency. Further longitudinal research is required to explore these associations among children from highly deprived areas. Future interventions may require a more holistic approach to improving FMS competency and increasing PA in order to account for the number of variables that can affect these outcomes.

## 1. Introduction

Early childhood is an important period for children to develop foundational movement skills (FMS) [1,2,3]. FMS is a recent new term that incorporates the established concept of *fundamental* movement skills, namely; stability (e.g., standing, balancing), locomotor (e.g., running, jumping), and object control (e.g., striking, catching, throwing) skills, alongside skills that can support lifelong engagement in physical activity (PA) (e.g., cycling, swimming) [1]. As such, developing competency in these skills during early childhood is important as FMS provide a base for successful participation in physical activity (PA) and sport across the life course [1,4,5,6]. However, despite the importance of developing FMS competency during early childhood, previous research has found low levels of FMS competency among preschool aged children [7,8]. 

Currently, the UK Government’s guidelines recommend that young children (up to five years of age) should aim to achieve 180 min of PA per day, with at least 60 min of this being moderate-to-vigorous PA (MVPA) [9]. However, there is a large body of evidence showing that preschool aged children spend a large amount of their day engaged in sedentary behaviour (SB) and a small proportion of their day in MVPA [9], with only one in 10 preschool children found to be meeting the recommended 180 min of MVPA per day. [10]. This is of particular concern given the important benefits of PA in young children relating to both their physical and psychological health [11]. Within the conceptual model developed by Stodden, Goodway et al. [2], it is hypothesized that the development of FMS competency is a primary underlying mechanism in promoting PA, with this association strengthening as children age. Furthermore, it is purported that this increase in PA brought about through increased FMS competency is able to shape positive or negative trajectories of weight status among children [2]. 

In 2015 a subsequent review paper by Robinson, Stodden [12] looked to synthesise the evidence relating to the previous Stodden, Goodway [2] model. This review summarized that there was strong evidence that FMS competency was positively associated with PA levels, cardiorespiratory fitness, muscular strength, muscular endurance and healthy weight status [12]. However, due to the large number of cross-sectional studies included in the review (only four studies were either longitudinal or experimental in design), this prevents any statements on causality from being made [13]. As such, further longitudinal studies are needed to examine the developmental trajectory of FMS competence with PA and obesity as proposed in the Stodden, Goodway [2] model. A recent systematic review by Barnett, Webster [14] looked to compile evidence from mediation, longitudinal and experimental studies in support of the Stodden, Goodway [2] model. Whilst the review found strong evidence for a negative association between FMS and weight status in both directions, there was indeterminate evidence for a pathway from FMS to PA and no evidence for the reverse pathway. Although there were findings of a bidirectional longitudinal associations between FMS competency and weight status, consistent with the Stodden, Goodway [2] model, the review authors recommended that further robust longitudinal and experimental studies examining changes in FMS relative to change in other constructs are needed [14]. 

Of the relatively limited longitudinal evidence that has examined these associations during childhood, particularly across the period of preschool to late primary (3–11 y), two previous studies found FMS competency to be a predictor of PA in children between primary age and adolescence [15,16].These findings would seem to suggest a causal relationship between FMS competency and PA. Barnett, Van Beurden [15] study among Australian children from 10.1–16.4 y. reported that adolescent time in MVPA was positively associated with childhood object-control competency, accounting for 12.7% (*p* < 0.05) of the variance. Furthermore, object-control proficient children were found to become adolescents with a 10% to 20% greater chance of participating in vigorous activity [15]. Lopes, Stodden [16] study observing Portuguese children from 6–14 y found a negative correlation (0.05–0.49) between BMI and motor coordination, measured using the Körperkoordinationstest Für Kinder [17]. 

Among preschool children, previous studies using accelerometers have reported positive but weak associations between FMS competency and PA [18,19,20,21,22]. Whilst Roscoe, James [23] cross-sectional study using accelerometers and the Test of Gross Motor Development-2 to assess 185 preschool children from a low socio-economic area within England found FMS competency did not appear to influence the level of PA or weight status in their sample. To the authors’ knowledge, no previous longitudinal study has examined the influence of FMS competency in relation to PA (using MVPA) among English preschool children. Likewise, longitudinal studies investigating the relationship between FMS competency and weight status are also required, specifically during early childhood, as evidence on associations between FMS competency and weight status among this age groups is limited. Longitudinal work by Bryant, James [24] has explored the relationship between FMS competency, PA and weight status among English primary school children, reporting that current FMS competency was a better predictor of current weight status, whilst prior FMS competence was a better predictor of current PA, however, PA data was obtained from pedometers. Zask, Barnett [25] longitudinal study looked at the effects of a movement skill intervention among preschool children on FMS competency and PA. After three years they reported no relationship between object-control skills and follow up MVPA, with the authors noting that this could have been due to a lack of MVPA data pre- and post-intervention to adjust for during analysis [25]. 

A study by Cohen, Morgan [26] examined the associations between FMS competency and PA among primary school children (8–9 y) living in low socioeconomic status (SES) areas. The authors noted that children living in low SES areas may be at greater risk of physical inactivity and other health inequalities [26], however, there is little longitudinal research regarding the relationship between FMS competency and PA among preschool children from deprived areas. Therefore, this study aims to examine the associations between FMS competency, objectively measured PA and weight status among preschool children attending preschools in highly deprived areas. Specifically looking at (i) how FMS competency and MVPA change with age (ii) how the association between FMS and MVPA changes with age and (iii) if preschool FMS competency is able to predict weight status at primary age. 

## 2. Materials and Methods

### 2.1. Study Design

This longitudinal study took place over a five-year period across 12 preschool/primary schools in Liverpool (a large urban city in Northwest England). Baseline assessments were conducted across two academic years, during October 2009 and March 2010. This approach was used to help maximise participant recruitment and minimise the influence of seasonal variation [27,28]. Follow-Up assessments were conducted between June and July 2015. Full ethical approval from the Research Ethics Committee at Liverpool John Moores University was received for data collection at Baseline (09/SPS/027) and Follow-Up (15/SPS/014). 

#### 2.1.1. Baseline

Baseline data for this study were drawn from the 2010 Active Play Project, which has been reported in detail elsewhere [7,29,30,31,32,33]. In brief, the Active Play Project was a Local Authority funded programme in response to a growing awareness of a need to establish health behaviours, such as increased PA, and promote fitness from an early age [34,35]. The project consisted of a six-week educational programme directed at preschool staff and children with the aim of increasing children’s PA levels, developing FMS, strength, agility, co-ordination and balance, and increasing children’s self-confidence. In line with the project’s funding requirements, each of the 12 preschools invited to take part in the study were selected in order to help address health inequities and improve indicators of child health. Such as, childhood obesity (12.2% of five-year-olds were obese) and PA levels that were significantly worse than the national average Association of Public Health [36]. Additionally, each preschool was attached to a Surestart children’s centre. The role of these centres is to offer advice, support and deliver services to parents and carers of children aged five years or under residing in the most deprived areas of England Children, Schools and Families [37]. All 12 of the preschools approached were situated within neighbourhoods ranked in the most deprived decile for deprivation nationally at the time of the study [38]. 

At the time of baseline assessments, all three- and four-year-old children in England were eligible to receive 15 h of fee-free preschool education for 38 weeks of the year. Four-year-old children were either attending under this offer or had recently commenced full time compulsory education (i.e., Monday to Friday, between the hours of 09:00 and 15:00). All 12 preschools agreed to take part, with all children aged 3–4.9 y in attendance at each preschool invited to participate (*n* = 673). Active consent was mandatory for those wishing to participate, with parents/carers providing informed written consent, demographic information (home postcode, child ethnicity and child’s date of birth) and completed medical assessment forms. All children were eligible to participate, however, children who, as identified by parental self-report, had been previously diagnosed with health or co-ordination issues that could affect their motor development, were excluded from subsequent analysis. In total, 240 children agreed to participate (mean age 4.5 y, ± 0.6 y; 51.7% male). 

#### 2.1.2. Follow Up

Each of the 12 preschools who participated in the 2010 study were situated within a primary school. As such, it was expected that the majority of preschool children would go on to attend the respective primary school. Due to the long period between Baseline and the proposed Follow-Up, researchers collaborated with the Senior School Improvement Officer (SSIO) from the Local Authority. This sped up the recruitment process, allowing for the identification of children who were now attending different primary schools (outside of the original 12) or had moved away entirely. The head teacher from each primary school was invited to attend a meeting with the SSIO and members of the research team. During this meeting the research team outlined the proposed study and what would be required should schools agree to take part, as well as answering any questions head teachers may have had. Head teachers who were unable to attend the meeting received an information pack outlining the project details. All 12 primary schools agreed to take part in the study, with gatekeeper consent obtained from each school’s head teacher. 

Of the 240 children who had participated in 2010, 181 children were identified as being in attendance across the 12 primary schools. All 181 children were invited to participate in the project and asked to return informed written parental consent and medical forms. In total, 131 children (mean age 10.0 y, ± 0.6 y; 52.3% male) agreed to participate in the study (72.4% response rate; 54.5% of original 2010 participants). Both children and their parents/carer were made aware that children were free to withdraw from the study at any point, without a need to provide a reason. 

### 2.2. Measures

#### 2.2.1. Foundational Movement Skills

FMS were examined using the Test of Gross Motor Development-2 (TGMD-2) [39] protocol. The TGMD-2 is specifically designed and validated for children aged 3–10 y, assessing six locomotor (run, broad jump, leap, hop, gallop and slide) and six object-control (overarm throw, stationary strike, kick, catch, underhand roll and stationary dribble) skills. A senior member of the research team with significant experience in administering the TGMD-2 was responsible for training all field testers, via in-situ observation, prior to the start of data collection at both time points. Children completed the TGMD-2 in small groups (2–4) in either school halls or outside on school playgrounds. Each group was led by two research assistants, the first provided a verbal description and single demonstration of each skill, whilst the second was responsible for recording each trial using a tripod mounted video camera. Children were required to perform each skill twice, with all 12 skills completed in the same order, taking approximately 35–40 min per group. 

Video recordings were subsequently converted to DVD, allowing for video analysis of each skill to be conducted at a later date. Skills were assessed using the Children’s Activity and Movement in Preschool Study Motor Skills Protocol (CMSP) [40]. The CMSP is a process-oriented assessment, with established validity and reliability [40], that evaluates each skill based on the demonstration of specific movement components [40]. Developed using an identical protocol to the TGMD-2 [39], the CMSP provides improved assessment sensitivity due to its additional performance criteria and alternative scoring methods [40]. For both trials of each skill, individual components, ranging in number from 3–8 dependent on the skill, were marked as being absent (scored 0) or present (1), with the exception of three skills. For the throw and strike, hip/trunk rotation was scored as *differentiated* (2), *block* (1) or *no rotation* (0), whilst the catch differentiated a successful attempt as being *caught cleanly with hands/fingers* (2) or *trapped against body/chest* (1). The total number of skill components checked as present over the two trials was summed to give a total FMS score. Additionally, object-control and locomotor scores were created by summing the number of present components within each subscale. 

All analyses were completed by a single trained assessor (JF), having received 30 h of training from a member of the research team experienced in undertaking video assessment of FMS (LF). Inter-rater reliability was established through the use of pre-coded DVDs of 10 children undertaking the TGMD-2 protocol, with an 83.9% agreement found across the 12 skills (range 72.9–89.3%) for the individual components of each skill. Despite there being no accepted minimum level of percentage agreement, 80–85% agreement has previously been deemed as acceptable [41]. Intra-rater reliability was further established using pre-coded DVDs of a further 10 children at baseline and follow up, with test-retest taking place one week apart. This resulted in agreement levels for the 12 skills of 91.9% (range 89–96%) for Baseline assessment and 97.0% (range 95–98.5%) for Follow-Up assessment. 

#### 2.2.2. Anthropometry

Body mass (to the nearest 0.1 kg) and stature (to the nearest 0.1 cm) were measured onsite by trained research assistants, using calibrated digital scales (Tanita WB100-MA, Tanita Europe, The Netherlands) and a portable stadiometer (Leicester Height Measure, SECA, Birmingham, UK), respectively. Body mass index (BMI, kg/m^2^) was calculated and converted to BMI-z scores using the “LMS” method for analysis [42].

#### 2.2.3. Physical Activity

PA levels were measured using hip-mounted uni-axial accelerometers ActiGraph GT1M and GT3X+ accelerometers, (ActiGraph, Pensacloa, FL, USA), with both models able to provide a similar level of validity in 5- to 9-year-old children [43]. Measures were taken every five seconds over a period of seven consecutive days. Children were asked to wear their accelerometer during all waking hours, with the exception of water-based activities e.g., bathing or swimming. Accelerometer data was reduced and analysed using ActiLife v6.0 (ActiGraph, Pensacloa, FL, USA). Valid wear time was defined as a minimum of three days, with at least nine hours of data recorded between 06:00 h and 23:59 h (waking hours). Non-wear time was defined as 20 min of consecutive zeros. Informed by the findings of Janssen, Cliff [44] age-appropriate cut points were used at the two time points. For Baseline SB data, Evenson, Catellier [45] cut-off points were used, whilst Pate, Almeida [46] cut-off points were used for Baseline MVPA. At Follow-Up only Evenson, Catellier [45] cut-off points were used. This approach of using differing cut points for PA over time was due to the Evenson, Catellier [45] cut-points having demonstrated significantly higher accuracy for classifying sedentary behaviors and light-intensity PA [44]. Additionally, the Evenson, Catellier [45] cut points have also been shown to provide acceptable classification accuracy for all four levels of PA intensity (sedentary, light, moderate and vigorous) and performed well among children of all ages [47]. PA data was categorised into average minutes of daily MVPA for subsequent analysis. To account for seasonal variation in data collection periods, the mean temperature (mean of daily minimum and maximum), rainfall (mm) and day length (sunrise to sunset; h) of each monitoring period was calculated for each participant. Daily temperature and day length data were obtained from www.timeanddate.com (accessed on 30 September 2021) and daily rainfall data from MET office records (http://www.metoffice.gov.uk/hadobs/hadukp/data/download.html) (accessed on 30 September 2021). 

### 2.3. Data Analysis 

Data were analysed using SPSS v23.0 (IBM Corporation, New York, NY, USA). For descriptive analysis, results are presented as means ± standard deviation and median and inter-quartile range for non-normally distributed data. A 2 (Baseline versus Follow-Up) × 2 (non-overweight versus overweight/obese) × 2 (boys versus girls) repeated measures ANCOVA was used to examine changes in FMS competency and PA with age, taking into account sex and weight classification differences. These were both adjusted for age, deprivation level [38], ethnicity, intervention/control classification, accelerometer wear time and seasonal PA variation (mean temperature, rainfall and day length). Participants from intervention and control groups were included in the present study as it was not important to differentiate between these two groups for this study, however, models did adjust for participant’s intervention status. Initially mixed linear models were run, adjusting for school level, however, school was found to have no effect. As such, linear models were used to examine if Baseline FMS competency predicted Follow-Up PA, whether Baseline PA predicted Follow-Up FMS competence and how the association between FMS competency and PA changed between Baseline and Follow-Up. Finally, binary logistic regressions were used to examine whether Baseline FMS competency predicted Follow-Up weight status (non-overweight or overweight/obese), respectively. Statistical significance was set at *p* < 0.05. Interactions by sex and weight status were explored but none were found (*p* < 0.10), thus regression models are presented at the group level. 

## 3. Results

In total, 75 children (58%) of the 131 whom provided full informed consent at Follow-Up (31% of original Active Play participants) met the inclusion criteria for this study (i.e., complete Baseline data and Follow-Up data for age, BMI, gender, PA data and total FMS score) and were subsequently included in the final analysis. There were no significant differences in Baseline characteristics between participants taking part in the present study and those not retained or excluded, except for deprivation score. A Mann-Whitney U test found that participants in the present study had a higher deprivation score (Md = 3.84, IQR = 1.01, 20.04, *n* = 75) than those excluded (Md = 2.79, 0.59, 4.85, *n* = 153). 

Table 1 shows participant characteristics in 2010 (M age 4.58 y ± 0.48; 50.7% boys; 29.7% overweight/obese; 85.1% White British; 84.0% lived in a low SES area) and in 2015 at Follow-Up (M age 9.98 y ± 0.49; 50.7% boys; 29.7% overweight/obese; 85.1% White British; 75.0% lived in a low SES area). There were significant increases (*p* < 0.05) in age, BMI, total, object-control, locomotor and deprivation scores between Baseline and Follow-Up, whilst MVPA (Baseline M = 90.3 ± 24.5; Follow-Up M = 69.0 ± 21.7) and monitor wear time (Baseline M = 779.5 ± 101.2; Follow-Up M = 695.4 ± 57.1) were found to significantly decrease (*p* < 0.05). Furthermore, there were significant (*p* < 0.05) differences in seasonal factors between Baseline and Follow-Up, with significant increases in daily temperature (Baseline M = 9.9 ± 1.0; Follow-Up M = 9.9 ± 1.0) and day length (Baseline M = 11.7 ± 0.9; Follow-Up M = 16.4 ± 1.2), and a significant decrease in rainfall (Baseline M = 3.3 ± 2.0; Follow-Up M = 1.5 ± 1.0).

At Baseline, the only significant sex difference was for object-control score, where boys scored significantly (*p* < 0.05) higher than girls. At Follow-Up significant differences (*p* < 0.05) were observed between boys and girls for total (boys M = 40.11 ± 5.01; girls M = 36.24 ± 4.75) and object-control (boys M = 19.45 ± 4.05; girls M = 15.03 ± 3.62) skill scores and MVPA (boys M = 78.56 ± 23.55; girls M = 59.25 ± 14.30). When looking at differences between Baseline and Follow-Up descriptives by sex, all changes were significant with the exception of BMIz score. 

### 3.1. How Does FMS Competency and MVPA Change with Age?

For descriptive purposes, Figure 1, Figure 2, Figure 3 and Figure 4 show the individual level and mean (± SD) changes in FMS scores and MVPA between Baseline and Follow-Up. There was an overall pattern of increase for total, object-control and locomotor scores between Baseline and Follow-Up (see Figure 1, Figure 2 and Figure 3). However, some differing trajectories were evident among participants: children who had lower scores at Baseline appeared to show greater levels of improvement to Follow-Up than their peers who had higher competency scores at Baseline. However, in general, competency scores were still found to be low, falling far short of the maximum attainable scores. Conversely, there was an overall pattern of decline for MVPA between Baseline and Follow-Up (see Figure 4).

Table 2 and Table 3 provide descriptive statistics alongside a summary of the repeated measures ANCOVA for all three FMS competency scores and MVPA. Table 2 shows that participants in both weight categories (non-overweight and overweight/obese) demonstrated an improvement in competency scores between Baseline and Follow-Up, with a main effect for time *p* < 0.05. Non-overweight participants had higher skill competency scores than their overweight/obese peers at both time points, although there was no significant effect for time × weight class (*p* > 0.05). For MVPA there was a significant effect for time × weight status (*p* < 0.05) between the two groups. Whilst both groups decreased their time spent in MVPA between Baseline and Follo-Up, there was a significantly greater decrease observed over time among overweight/obese children, with overweight/obese children spending less time in MVPA at follow up.

Table 3 reports the differences in competency scores and MVPA over time by sex. Both boys and girls significantly improved their competency scores between Baseline and Follow-Up, with a main effect for time (*p* < 0.05). There was a significant time × sex interaction (*p* < 0.05) for total and object-control scores, with boys having been found to have significantly greater increases in total and object-control scores between Baseline and Follow-Up in comparison to girls. Boys spent more time than girls in MVPA at both time points, although this did not result in any significant differences. Both sexes spent significantly less time in MVPA at follow up compared to Baseline (*p* < 0.05). No significant interactions were found. 

### 3.2. Does Baseline FMS Competence Predict MVPA at Follow Up?

Results of the regression analyses examining FMS competency scores as predictors of MVPA at Follow-Up are summarised in Table 4. Having controlled for intervention group alongside Baseline age, deprivation, ethnicity, sex, BMI-z and Follow-Up monitor wear time and weather, none of the FMS competency scores were found to be significant predictors of MVPA at Follow-Up (*p* > 0.05). Total, locomotor and object-control skill score at Baseline each predicted less than 1% of unique variance in MVPA at Follow-Up.

### 3.3. Does Baseline MVPA Predict FMS Competence at Follow-Up?

Outcomes from regression analysing whether Baseline MVPA predicted FMS competence at Follow-Up are presented in Table 4. Having controlled for Baseline monitor wear time and weather in addition to the stated covariates, Baseline MVPA did not significantly predict FMS competency at Follow-Up (see Table 4). Baseline MVPA predicted only 2% of unique variance in total FMS score. When sub-domains of FMS were examined, baseline MVPA predicted only 2% of unique variance in object-control competency score and 0.2% of unique variance in locomotor skill score. 

### 3.4. How Does the Association between FMS and MVPA Change between Baseline and Follow-Up?

The strength of association between FMS and MVPA at Baseline is shown in Table 4. After adjustments, total and locomotor skill scores significantly predicted MVPA (*p* < 0.05). Total FMS score predicted 4.5% of unique variance in Baseline MVPA; specifically, a one unit increase in total skill score was associated with a 1.04 min increase in Baseline MVPA (95% CI, 0.20 to 1.9). When looking at the sub-domains, locomotor skill score predicted 3.3% of unique variance, with a one unit increase in locomotor score accounting for a 1.3 min increase in MVPA (95% CI, 0.06 to 2.61). However, object-control score was not found to be a significant predictor (*p* > 0.01), accounting for only 0.5% of unique variance in baseline MVPA. 

When looking at the relationship between FMS and MVPA at Follow-Up, a further regression controlling for stated covariates, found none of the three competency scores to be significant predictors of MVPA (*p >* 0.01) (see Table 4), indicating that the strength of association between FMS and MVPA weakened over time. At follow up, total and locomotor scores had decreased in their level of prediction of unique variance to 2.5% and 0.5%, respectively. Whilst there was an increase in object-control score compared to Baseline, now predicting 1.6% of unique variance in MVPA, this was not significant (*p* > 0.01). 

### 3.5. Does Baseline FMS Competency Predict Follow-Up Weight Classification?

The results of the binary logistic regression (see Table 5) show that (controlling for intervention group and Baseline age, deprivation, ethnicity, sex, BMI-z score, monitor wear time and weather) none of the three skill competency scores significantly predicted Follow-Up weight classification (i.e., non-overweight or overweight/obese). 

## 4. Discussion

To the authors’ knowledge, this is the first longitudinal study to examine the relationship between FMS competency, PA and weight status among English children living in areas of high deprivation throughout the period of early to late childhood. The main findings from the present study were that despite significant increases from Baseline to Follow-Up, FMS competency was still low among participants at Follow-Up. Boys had significantly higher total and object-control scores than girls at Follow-Up, whilst overweight/obese (OW/OB) children had lower competency levels than their non-overweight (NW) peers for all competency scores at both Baseline and Follow-Up. There was a significant decrease in MVPA among both boys and girls between Baseline and Follow-Up, with a significantly greater decrease observed over time among OW/OB children. Associations between FMS competency and MVPA were found to be weak, with Baseline FMS competency and MVPA failing to significantly predict Follow-Up levels of MVPA and FMS competency, respectively. Likewise, Baseline FMS competency was not found to be a significant predictor of child weight status at Follow-Up. Furthermore, the association between FMS competency and MVPA was found to weaken over time; at Baseline, total and locomotor skill competency scores were significant but weak predictors of MVPA. However, at Follow-Up skill competence did not predict MVPA.

Participants in the present study were found to have low FMS competency at Baseline, with significant (*p* < 0.05) increases in total, object-control and locomotor competency scores observed at Follow-Up. Similarly, Butterfield, Angell [48] noted a rapid increase in children’s competency levels between the ages of 5–10 y, with low competency levels expected prior to 5y. Although, it is possible that these differences in FMS competency across time points could be due to a relative age effect [49]. However, despite the significant increase in competency scores among both boys and girls in the present study, competency levels at Follow-Up were still low. With previous studies having found that low FMS competency tracks over time [50,51], these results are perhaps not surprising. Likewise, at Follow-Up, boys had significantly higher total and object-control competency scores than girls. This is in line with previous studies where boys have been found to be more competent at object-control skills [52,53,54,55]. Most notably, Bryant, Duncan [53] study among 281 English primary school children (M age 8.4 ± 1.6 y) reported boys to be more competent at kicking and catching, alongside overall low levels of competency among the children observed. 

The findings of low competency in the present study may be related to the participants residing in areas of high deprivation, with previous studies among children from highly deprived areas reporting low levels of competency [7]. Goodway, Robinson [56] noted that among 469 American preschool children from highly deprived areas, children had low competency in object-control and locomotor skills. Furthermore, the authors found boys to be significantly more competent at object-control skills. Likewise, Morley, Till [57] found low competency levels for motor proficiency among their sample of 369 low SES English children (age 4.3–7.2 y). Participants were found to have significantly lower motor proficiency in comparison to socioeconomically advantaged children, whilst boys within the study were found to have outperformed girls for the object-control skills, catch and dribble [57]. A previous qualitative study by Goodway and Smith [58] highlighted the issues of a lack of access to safe outdoor play, the availability of neighbourhood or family resources to access equipment and/or youth sports and limited physical activity role models as barriers to PA among disadvantaged children. If, therefore, children residing in low SES areas have less opportunities to engage in PA, this may result in them having fewer chances to practice FMS and thus lead to lower competency levels. As such, children from low SES areas may require more instruction and practice of FMS in order for them to achieve similarly high levels of competency as their peers from areas of low deprivation. However, to be successful, interventions may also need to take into account sex differences alongside SES status [57]. With previous findings of boys outperforming girls for object-control skills [7,59,60], interventions may need to ensure that girls are not disadvantaged in activities requiring object-control skills, whilst also ensuring that boys displaying high competency levels receive sufficient opportunities to continue developing their competency levels [56]. 

According to Robinson, L.E.; et al. [12] there is a reciprocal and developmentally dynamic relationship between FMS competency and PA during childhood, which should strengthen over time between early and late childhood. In the present study a significant positive association (*p* < 0.05) was found between FMS competency and PA at Baseline, with total and locomotor competency scores found to predict MVPA. These observed positive associations between competency scores and PA fall in line with previous studies examining associations between competency and PA among young children [18,19,20,29]. Likewise, the current study’s findings of boys engaging in more MVPA and displaying greater object-control skill competency is consistent with the literature [32,59,60,61,62]. However, in direct contrast to the Stodden, Goodway [2] model, associations between FMS competency and MVPA were found to have weakened over time, with no significant associations found between competency scores and MVPA at Follow-Up. Whilst there was a small increase in the unique variance in MVPA accounted for by object-control score between Baseline and Follow-Up, this was still relatively low (1.56%) and was not significant (*p* > 0.05).

It is possible that this weakening of association could in part be due to the large decrease in time spent in MVPA observed between Baseline and Follow-Up among participants, with declines in MVPA having previously been reported across childhood [63]. This decrease in MVPA was especially true for OW/OB children, whereby a significant effect for time was observed, noting that there was a significantly greater change in their time spent in MVPA compared to their NW peers. Similar to the present study Cohen, Morgan [26] examined the association between FMS competency and MVPA, using a process-based measure of FMS (TGMD-2) and accelerometry, among 460 low SES Australian primary school children (M 8.5 ± 0.6 y). Whilst their study found significant associations between locomotor and object-control scores and MVPA, their analysis did not control for covariates such as monitor wear time or weather conditions, as the present study has.

Stodden, Goodway [2] note that further research examining this relationship should take into account mediating variables that may interact with and promote/demote the dynamic relationship between FMS competency and PA within their model. Factors such as SES [7,56] or parental/carer influence [64,65] are not included in the model [2], but may influence and weaken the relationship between FMS competency and PA. Furthermore, Sterdt, Liersch [66] recently conducted a systematic review and identified 16 correlates that were consistently associated with PA in children and adolescents. This highlights that PA is a complex and multi-dimensional behaviour, determined by numerous biological, psychological, sociocultural and environmental factors. As such, a more holistic model of motor competence may be needed in order to account for the large number of variables that can affect physical activity participation over time.

Looking at the early years as an important phase for FMS development and PA behaviours, the present study failed to find an association between Baseline FMS competency as a predictor of Follow-Up MVPA, or Baseline MVPA as a predictor of Follow-Up FMS competency. Whilst Bryant, James [24] longitudinal study among English primary school children reported that prior FMS competency was a better predictor of current PA, their study only measured follow up data after one year. Consequently, there might not have been as great a change in competency/PA levels as in the present study’s five-year timeframe. Bryant, James [24] study also used pedometers to record PA and as such could only report on associations between FMS competency and total PA, without being able to identify any associations between FMS competency and specific bouts of PA. Lopes, Rodrigues [67] reported that among 6–10 y old children that FMS competency was an important predictor of PA. Similarly, Barnett, Van Beurden [15] found that children with object-control competency at late primary were more likely to be active in adolescence. However, both of these studies assessed PA using questionnaires, deemed less reliable to that of accelerometer data due to the potential issue of recall errors among participants, especially among children [68]. 

In the present study OW/OB children were found to have lower competency across all scores at both time points in comparison to NW children. These findings are in line with previous studies that have reported that BMI is negatively associated with FMS competency [16,20,54,69,70,71]. An increasing difference in FMS competency between OW/OB and NW children across ages has been documented in a previous cross-sectional study by D’Hondt, Deforche [13], with NW children showing greater competency levels. In a more recent longitudinal study by D’Hondt, Deforche [72] the authors reported a widening gap between OW/OB children’s FMS competency relative to their gender and age matched NW peers. The authors believed that this increasing difference in competency between OW/OB and NW children was mainly attributable to NW children showing greater improvements in competency over the short term, in comparison to their OW/OB peers. However, previous studies among OW/OB children have reported that interventions incorporating regular PA as a central component resulted in short-term improvements in motor competency [73,74], indicating that it is possible for OW/OB children to increase their competency levels and narrow the competency gap between themselves and NW children. As such, it may be possible to reduce the competency gap between NW and OW/OB children in the present study. To do so, appropriate interventions would be required that address the deficiencies in FMS competency of OW/OB children, allowing these children to develop the required movement skills to engage in regular, health-enhancing physical activity [75]. This was further highlighted in a systematic review of 17 studies by Han, Fu [76] looking at the effectiveness of exercise/PA interventions on improving FMS competency among OW/OB children and adolescents. The review concluded that interventions aimed at improving FMS competency among OW/OB children and adolescents should be enjoyable and specifically targeted at increasing FMS competency [76]. 

Finally, the present study found no association between Baseline competency scores and Follow-Up weight classification. As such, no inference could be made to support the Stodden, Goodway [2] model hypothesis that the development of FMS competency is a primary underlying mechanism in promoting PA and therefore shaping positive or negative trajectories of weight status among children. The findings of previous studies examining the association between FMS competency and weight status support Stodden, Goodway [2] assertion that FMS competency is both a *precursor* and *consequence* of childhood weight status. Okely, Booth [77] examined the association between FMS competency and BMI among 4363 Australian children and adolescents (9–16 y). The results indicated that OW children of both sexes were less likely to have high levels of FMS competency, with FMS competency further found to be significantly related to BMI. For object-control and locomotor competency, NW boys and girls were two to three times respectively more likely to possess more advanced locomotor skills than their OW peers. These findings may indicate that interventions aiming to prevent weight gain among children may benefit from focusing on increasing locomotor skill competency [77].

The main strength of the present study was the use of a validated process-based measure of FMS competency [40], via video analysis by a single trained assessor, providing confidence and consistency in the measurement of children’s competency levels. Furthermore, the use of accelerometers allowed the opportunity to objectively assess participants PA. However, in using cut points to determine absolute PA intensity and PA intensity relative to the individual, there is the possibility that there will be miscalculations in the data due to cut points not accounting for relative intensity differences and should be considered as a potential limitation. A further limitation of the present study was the 58% participation rate of children approached to participate (*n* = 131), which only accounted for 31% of the original Active Play participants. This highlights the difficulties of trying to collect data from participants as part of a large-scale longitudinal study. Furthermore, the use of accelerometers to obtain PA data means that water-based or non-ambulatory activities cannot be recorded and so MVPA may have been underestimated.

## 5. Conclusions

This is the first longitudinal study to examine the associations between FMS competency, PA and weight status among English children. Despite the lack of significant associations found between FMS competency and MVPA, these findings can contribute to that of the current literature. Firstly, the failure to find a strengthening association between FMS competency and MVPA over time contradicts the proposed Stodden, Goodway [2] model of a reciprocal and dynamic relationship between FMS competency and PA. Secondly, low levels of competency at Baseline and Follow-Up and a significant decline in MVPA among children in this study draws attention to the need to intervene in this age group. Further longitudinal research is therefore required to continue to examine the associations between FMS competency and MVPA, among both high and low SES preschool children, allowing for additional comparisons to be made between these differing groups. The weak associations found in the present study also show how large amounts of variance in MVPA are explained by a number of different variables, outside of those measured in the present study, or put forward in the Stodden, Goodway [2] model and subsequent Robinson, Stodden [12] review paper. As such, this would indicate that more holistic interventions may be required, controlling for as many external variables as possible e.g., children’s motivation and confidence, in order to promote sustained participation in PA. 

## Figures and Tables

**Figure 1 ijerph-19-00105-f001:**
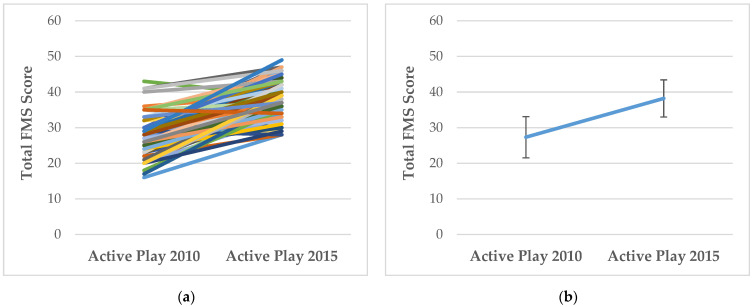
(**a**) Individual changes in total FMS scores between Baseline (Active Play 2010) and Follow-Up (Active Play 2015); (**b**) Mean (±SD) changes in total FMS scores between Baseline (Active Play 2010) and Follow-Up (Active Play 2015).

**Figure 2 ijerph-19-00105-f002:**
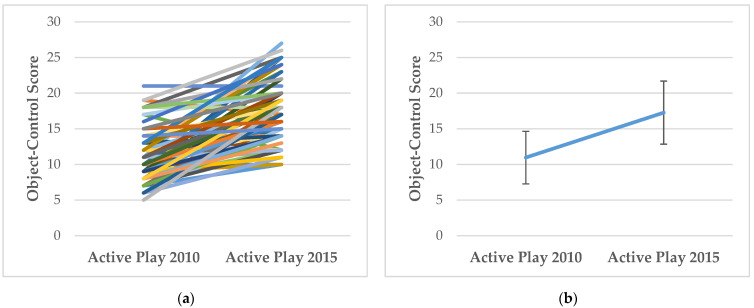
(**a**) Individual changes in object-control scores between Baseline (Active Play 2010) and Follow-Up (Active Play 2015); (**b**) Mean (±SD) changes in object-control scores between Baseline (Active Play 2010) and Follow-Up (Active Play 2015).

**Figure 3 ijerph-19-00105-f003:**
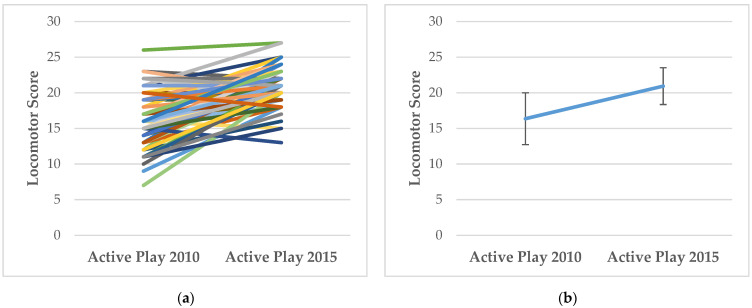
(**a**) Individual changes in locomotor scores between Baseline (Active Play 2010) and Follow-Up (Active Play 2015); (**b**) Mean (±SD) changes in locomotor scores between Baseline (Active Play 2010) and Follow-Up (Active Play 2015).

**Figure 4 ijerph-19-00105-f004:**
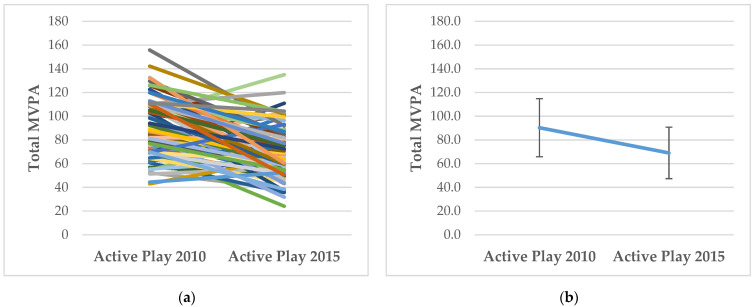
(**a**) Individual changes in MVPA between Baseline (Active Play 2010) and Follow-Up (Active Play 2015); (**b**) Mean (±SD) changes in MVPA between Baseline (Active Play 2010) and Follow-Up (Active Play 2015).

**Table 1 ijerph-19-00105-t001:** Baseline (Active Play 2010) and Follow-Up (2015) descriptive characteristics for participants (Mean ± SD; Median and inter-quartile range).

	Active Play 2010			Active Play 2015		
Measure	Boys	Girls	Total	Boys	Girls	Total
	(*n* = 38)	(*n*= 37)	(*n* = 75)	(*n* = 38)	(*n* = 37)	(*n* = 75)
Age (y)	4.5 ± 0.6	4.6 ± 0.4	4.6 ±0.5	10.0 ± 0.6 *	10.0 ± 0.4 *	10.0 ± 0.5 *
BMI (kg/m^2^)	16.7 ± 1.8	16.6 ± 1.9	16.7 ± 1.8	18.5 ± 3.4 *	18.7 ± 3.6 *	18.6 ± 3.5 *
BMI-z Score (IOTF)	0.7 ± 1.1	0.7 ± 1.1	0.7 ± 1.1	0.7 ± 1.1	0.7 ± 1.1	0.7 ± 1.1
MVPA (min)	95.6 ± 22.8	84.9 ± 25.3	90.3 ± 24.5	78.6 ± 23.6 *	59.3 ± 14.3 *	69.0 ± 21.7 *
Wear Time (min)	780.3 ± 105.7	778.6 ± 97.9	779.5 ± 101.2	698.5 ± 57.3 *	692.2 ± 57.4 *	695.4 ± 57.1 *
Total FMS ‡	28.2 ± 5.9	26.41 ± 5.6	27.3 ± 5.8	40.1 ± 5.0 *	36.2 ± 4.8 *	38.2 ± 5.2 *
OC Score ‡	12.3 ± 3.8	9.5 ± 3.0	11.0 ± 3.7	19.5 ± 4.1 *	15.0 ± 3.6 *	17.3 ± 4.4 *
LM Score ‡	15.9 ± 3.7	16.9 ± 3.6	16.4 3.6	20.7 ± 2.3 *	21.2 ± 2.9 *	20.9 ± 2.6 *
Temperature (°C)	9.8 ±1.1	10.1 ± 1.0	9.9 ± 1.0	15.5 ± 1.5 *	15.4 ± 1.3 *	15.4 ± 1.4 *
Rainfall (mm)	3.3 ± 2.0	3.2 ± 2.1	3.3 ± 2.0	1.5 ± 0.8 *	1.5 ± 1.1 *	1.5 ± 1.0 *
Daylength (hours)	11.7 ± 0.9	11.6 ± 1.0	11.7 ± 0.9	16.4 ± 1.1 *	16.3 ± 1.3 *	16.4 ± 1.2 *
**Median (IQR)**						
Deprivation †	4.03 (1.1, 20.7)	3.4 (0.8, 19.8)	3.84 (1.0, 20.4)	47.0 (21.0, 59.5)*	53.0 (31.8, 59.0) *	50.5 (24.8, 59.0) *

Note: OC, object-control; LM, locomotor; IOTF, International Obesity Task Force age- and sex-specific weight for height z scores. ‡ Maximum attainable score: Total FMS score 73; object-control skill score 39; and locomotor skill score 34. † Deprivation rank score. * Significantly different from baseline value (*p* < 0.05).

**Table 2 ijerph-19-00105-t002:** Means, standard deviations and summary of repeated measures analysis for FMS competency scores and MVPA for non-overweight and overweight/obese participants.

	Baseline (2010)		Follow-Up (2015)		Repeated Measures ANCOVA			
Score	NW	OW/OB	NW	OW/OB	F_Time_	*p*	F_Time_ × _Weight Class_	*p*
	(*n* = 52)	(*n*= 21)	(*n* = 52)	(*n* = 21)				
**FMS Score**								
Total	27.65 ± 5.96	25.82 ± 5.10	38.73 ± 5.21	36.33 ± 6.97	21.85	<0.001 *	0.000	0.99
OC	11.31 ± 3.65	9.62 ± 3.34	17.58 ± 4.30	15.90 ± 6.08	21.33	<0.001 *	0.18	0.67
LM	16.35 ± 3.77	16.24 ± 3.49	21.15 ± 2.52	20.43 ± 2.79	2.82	0.01 *	0.30	0.59
**MVPA** ^1^								
MVPA	88.33 ± 23.72	93.19 ± 26.91	71.40 ± 22.00	62.55 ± 21.11	1.31	0.26	4.95	0.03 *

Note: OC, object-control; LM, locomotor; NW, non-overweight; OW/OB, overweight/obese. All analyses corrected for age, deprivation score, ethnicity and participation in intervention group. ^1^ Further adjusted for weather and monitor wear time. * Significant at *p* < 0.05.

**Table 3 ijerph-19-00105-t003:** Means, standard deviations and summary of repeated measures analysis for FMS competency scores and MVPA among boys and girls.

	Baseline (2010)		Follow-Up (2015)		Repeated Measures ANCOVA			
Score	Boys	Girls	Boys	Girls	F_Time_	*p*	F_Time_ × _Weight Class_	*p*
	(*n* = 38)	(*n*= 37)	(*n* = 38)	(*n* = 37)				
**FMS Score**								
Total	28.21 ± 5.92	26.41 ± 5.57	40.11 ± 5.01	36.24 ± 4.75	37.89	<0.001 *	7.53	0.007 *
OC	12.34 ± 3.82	9.54 ± 2.98	19.45 ± 4.05	15.03 ± 3.62	26.43	<0.001 *	5.71	0.02 *
LM	15.87 ± 3.66	16.86 ± 3.58	20.66 ± 2.25	21.22 ± 2.89	15.46	<0.001 *	1.72	0.19
**MVPA** ^1^								
MVPA	95.57 ± 22.83	84.87 ± 25.27	78.56 ± 23.55	59.25 ± 14.30	13.86	<0.001 *	1.74	0.19

Note: OC, object-control; LM, locomotor; NW, non-overweight; OW/OB, overweight/obese. All analyses corrected for age, deprivation score, ethnicity, participation in intervention group. ^1^ Further adjusted for weather and monitor wear time. * Significant at *p* < 0.05.

**Table 4 ijerph-19-00105-t004:** Results from linear regression examining associations between FMS scores and MVPA.

Predictor	β	SE	95% CI	*p*	r^2^	sr_i_^2^
**Baseline FMS and Baseline MVPA ^1^**						
Total	1.04	0.42	0.20 to 1.9	0.02 *	53.8%	4.53%
OC	0.62	0.78	−0.94 to 2.19	0.43	54.1%	0.48%
LM	1.34	0.64	0.06 to 2.61	0.04 *		3.31%
**Baseline FMS and Follow-up MVPA ^2^**						
Total	0.08	0.39	−0.69 to 0.85	0.83	47.6%	0.04%
OC	0.30	0.70	−1.38 to 1.43	0.97	47.6%	<0.01%
LM	0.13	0.65	−1.16 to 1.43	0.84		0.04%
**Baseline MVPA and Follow-up FMS ^3^**						
Total	0.04	0.03	−0.20 to 0.11	0.19	26.6%	1.96%
OC	0.04	0.03	−0.01 to 0.09	0.15	39.6%	2.04%
LM	0.01	0.02	−0.03 to 0.04	0.69	18.2%	0.21%
**Follow-up FMS and Follow-up MVPA ^4^**						
Total	0.71	0.41	−0.12 to 1.54	0.09	56.0%	2.46%
OC	0.75	0.55	−0.35 to 1.86	0.08	56.0%	1.56%
LM	0.63	0.83	−1.04 to 2.30	0.16		0.49%

Note: β, unstandardized regression coefficient; SE, standard error for β coefficient; 95% CI, confidence intervals for regression coefficient; r^2^ total variance explained by Baseline score and predictor variables; sr_i_^2^, squared semi-partial correlation coefficient, unique variance explained by Baseline score; OC, object-control; LM, locomotor. All models adjusted for intervention group and Baseline age, deprivation, ethnicity, sex, and BMI-z score; ^1^ model additionally adjusted for Baseline monitor wear time and weather; ^2^ additionally adjusted for Follow-Up monitor wear time and weather; ^3^ additionally adjusted for Baseline monitor wear time and weather; ^4^ additionally adjusted for Follow-Up monitor wear time and weather. * Significant at *p <* 0.05.

**Table 5 ijerph-19-00105-t005:** Logistic regression of Baseline FMS competency predicting the likelihood of being overweight/obese at Follow-Up.

Predictor	B	SE	95% CI	Odds Ratio	*p*
Total	0.15	0.19	0.81 to 1.68	1.17	0.41
OC	−1.95	3.00	0.00 to 50.97	0.14	0.52
LM	1.42	2.04	0.08 to 225.57	4.12	0.49

Note: β, regression coefficient; SE, standard error; OR, adjusted odds ratio; 90% CI, confidence intervals. All models adjusted for intervention group and baseline age, deprivation, ethnicity, sex, BMI-z score, monitor wear time and weather.

## Data Availability

The data presented in this study are available on request from the corresponding author. The data are not publicly available in accordance with consent provided by the study participants.

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
