# Peer review of "Is Foundational Movement Skill Competency Important for Keeping Children Physically Active and at a Healthy Weight?"

_ijerph, 2021, doi:10.3390/ijerph19010105_

Round 1

Reviewer 1 Report

This paper excellent addition to the literature in the field of exercise science. The paper “Is Foundational Movement Skill Competency Important for Keeping Children Physically Active and at a Healthy Weight?” examines the association between foundational movement skills (FMS) competency, moderate-to-vigorous physical activity (MVPA), and weight status among children. Although, this paper has limitations its presents important conclusions.

Minor points to consider:

Previous studies using accelerometers have reported overestimate data, how do the authors compensate for this disadvantage using accelerometers?

Why do authors use the word “Foundational “instead of “fundamental”?

The authors introduce variables like biological, psychological, socio-cultural, and environmental factors, but none of them were determined in this study.

What is the PA activity that would be most beneficial to child development?

What are the long terms effects of foundational movement skills on Children's PA engagement?

Do the authors find any activity or play a game that motivates child movement skills?

What are the negative factors that affect foundational movement skills?

How to increase PA? is there any educational model?

Reviewer 2 Report

A well conceived and well designed study. It would be useful to articulate clinical practice application of the findings of the study.

Reviewer 3 Report

The work entitled: “Is Foundational Movement Skill Competency Important for Keeping Children Physically Active and at a Healthy Weight? ”, is a very interesting study that illustrates the importance of healthy lifestyle.

Overall, this manuscript is original and of great importance to the academic community and readers in general. However, some aspects of formatting, methodology and results must be revised to support the results.

Below are the comments referring to each section of the article:

  1. Methods: 155-6 and 171-7 , Include in results. 
  2. Use WHO curve classification for BMI.
  3. I found the description of the variable physical activity with many cutoff points confusing. 
  4.  
